# Sleep-Disordered Breathing among Saudi Primary School Children: Incidence and Risk Factors

**DOI:** 10.3390/healthcare11050747

**Published:** 2023-03-03

**Authors:** Saleh H. Alwadei, Suliman Alsaeed, Ahmed Ibrahim Masoud, Farhan Alwadei, Khalid Gufran, Abdurahman Alwadei

**Affiliations:** 1Department of Preventive Dental Sciences, College of Dentistry, Prince Sattam Bin Abdulaziz University, Alkharj 11942, Saudi Arabia; 2Preventive Dental Sciences Department, College of Dentistry, King Saud Bin Abdulaziz University for Health Sciences, Riyadh 11481, Saudi Arabia; 3King Abdullah International Medical Research Center, Riyadh 11481, Saudi Arabia; 4Ministry of the National Guard Health Affairs, Riyadh 11426, Saudi Arabia; 5Department of Orthodontics, Faculty of Dentistry, King Abdulaziz University, Jeddah 21589, Saudi Arabia; 6Department of Pediatric Dentistry and Orthodontics, College of Dentistry, King Saud University, Riyadh 11545, Saudi Arabia

**Keywords:** SDB, pediatric sleep questionnaire, sleep apnea, prevalence

## Abstract

This study aimed to identify the incidence and risk factors of sleep-disordered breathing (SDB) using an Arabic version of the pediatric sleep questionnaire (PSQ). A total of 2000 PSQs were circulated to children aged 6–12 years who were randomly selected from 20 schools in Al-Kharj city, Saudi Arabia. The questionnaires were filled out by the parents of participating children. The participants were further divided into two groups (younger group: 6–9 years and older group: 10–12 years). Out of 2000 questionnaires, 1866 were completed and analyzed (93.3% response rate), of which 44.2% were from the younger group and 55.8% were from the older group. Among all the participants, a total of 1027 participants were female (55%) and 839 were male (45%) with a mean age of 9.67 ± 1.78 years. It showed that 13% of children were suffering from a high risk of SDB. Chi-square test and logistic regression analyses within this study cohort showed a significant association between SDB symptoms (habitual snoring; witnessed apnea; mouth breathing; being overweight; and bedwetting) and risk of developing SDB. In conclusion: habitual snoring; witnessed apnea; mouth breathing; being overweight; and bedwetting strongly contribute the to development of SDB.

## 1. Introduction

Many airway dysfunctions, including obstructive sleep apnea (OSA) and primary snoring, are caused by sleep-disordered breathing (SDB) [1]. OSA is a severe form of SDB characterized by upper airway obstruction, either partial or intermittent, that interrupts the normal sleeping pattern [2]. Among children, SDB is linked with growth retardation, behavioral problems, disturbances in cognitive development, failure to thrive, and attention deficit/hyperactivity disorder [2,3]. The myriad of SDB symptoms in children also include abnormal breathing, snoring, sweating, aggressiveness, irritability, hyperactivity, sleepiness, excessive fatigue, memory impairments, and poor school performance, among many others [1,2,3,4,5]. Furthermore, the craniofacial morphology associated with SDB in children includes retrognathic mandible, increased lower facial height, narrow maxillary arch with a high vault, posterior crossbite, anterior open bite, and restriction in the upper airway space [5,6].

The prevalence of pediatric OSA ranges from 1–6%, while habitual snoring varies by definition and ranges from 4–17% [1,7,8,9,10]. A previous study in southern Italy found that 4.9% school children were possessed habitual snoring and among them only 1% had OSA [11]. Another Turkish study stated that the prevalence of 6–13 years children’s snoring habit is 7% [12]. Moreover, 11.4%, 12% and 27.6% prevalence rates of snoring habit were observed in the Indian, Chinese and Brazilian populations, respectively [13,14,15].

In Saudi Arabia, the prevalence among adults and children leans toward the higher end of the range reported in the literature [16,17,18]. However, the majority of children with OSA remain undiagnosed [19]. In order to diagnose OSA in children, nocturnal sleep-based polysomnography (PSG) is widely used and is considered the gold standard to diagnose OSA [8]. Regardless of its diagnostic advantages, PSG presents challenges related to its cost, time, complexity, and availability/access which necessitates the use of an efficient screening test that may prompt early detection and treatment [9,19,20]. As dentists and selected dental specialists obtain radiographic x-rays regularly and examine children daily, they could effectively identify the children who are at risk of SDB [9,21]. This is especially true given the previously mentioned association between SDB and craniofacial growth and development, as well as the proven positive responses to various treatment modalities ranging from interceptive orthodontics to orthognathic surgery [22,23,24].

Although thorough clinical history and physical examination are an integral part of best practice in any healthcare profession including assessment of SDB, they may not be sufficient to identify children suspected (or at high risk) of OSA [19]. It is important to consider efficient, reliable, and accurate screening tests for timely diagnosis and management (including referral) of children with SDB since it might prevent associated comorbidities [20]. Pediatric sleep questionnaires have been developed for the screening of SDB in children and adolescents, and epidemiological studies have shown them to be clinically significant and relevant [25,26]. As subjective parent report tools, sleep questionnaires are apprehensive about the risk factors and sign symptoms of SDB and OSA [27]. Among these questionnaires, the pediatric sleep questionnaire-22 (PSQ-22) has been validated in groups of referred snoring children and controls, showing excellent specificity and sensitivity for identifying children with OSA [2]. Therefore, this study aimed to determine the incidence of SDB and identify related risk factors among primary school children in Al-Kharj, Saudi Arabia.

## 2. Materials and Methods

This was a cross-sectional study that was conducted from September 2018 to December 2018 in Al-Kharj city, Saudi Arabia. Al-Kharj city is situated in the Riyadh province of Saudi Arabia with about 376,325 residents (https://www.stats.gov.sa (accessed on 21 November 2017)). Ethical approval was obtained from the College of Dentistry Research Centre at Prince Sattam Bin Abdulaziz College of Dentistry (Registration No: 1439-03-003). In addition, permission to conduct this research was also obtained from the authority of the Ministry of Education. A stratified randomization technique was used to list the schools to be included in this study to ensure the proper sample representation in the Al-Kharj. A free online randomization software (http://www.randomization.com (accessed on 21 November 2017)) was used to select 20 schools (10 boys’ schools and 10 girls’ schools). A formal letter was sent to the principal of each school to obtain permission by explaining the purpose of the study. Once the school authorities agreed to participate in the current research, they were requested to provide a list of primary school students with an assigned number corresponding to each student. Then, a randomization table was used to select the students.

A required sample size of 454 subjects was estimated based on a statistical power calculation described by Pourhoseingholi and colleagues (2013) [28], considering 0.20 non-responses, a confidence level of 95%, a precision of 0.04, and 20% prevalence SBD as reported in Saudi-based literature using PSQ [17,18].

Each child was given a folder in order to obtain permission from the parents which included: (1) a cover page explaining the aim of the study, the significance of the study, and the confidentiality measures taken to protect collected information, (2) a consent form, and (3) the PSQ. Parents/guardians were asked to observe their child’s sleep pattern for one week before filling out the PSQ to improve response accuracy. Children aged 6–12 years who presented a consent form signed by their parents were included in the current study. The participants were divided into two groups by age: the younger group (age 6–9 years) and the older group (age 10–12 years).

A total of 2000 folders containing a cover letter, a consent form, and the PSQ were distributed to randomly selected children, and the folders were collected a week later.

The PSQ was previously validated by Chervin et al. (2000); therefore, no validation was required for this study. Moreover, the Arabic version of PSQ was validated by Baidas et al. (2019); therefore, the Arabic form of PSQ was used in this study in order to explain the PSQ properly by the parents [17]. The PSQ contains 22 items, and each item consists of three options to respond to with the following options: ‘yes’ = 1; ‘no’ = 0; and ‘don’t know’ = missing. If participants scored ≥8 items to ‘yes’, they would be considered at high risk of SDB, whilst if they scored <8 items to ‘yes’, they would be considered at a low risk of SDB.

### Statistical Analysis

Statistical analysis was performed using SPSS software version 28 (IBM Corp. Armonk, NY, USA). The demographic of the children and the prevalence rate of SDB was assessed with descriptive statistics. A Chi-square test was performed to identify the differences in demographic variables and SDB symptoms related to the risk factors. For the Chi-square test (2 × 2), demographic variables were dichotomized as (male/female) for gender and (older/younger) for age, when the test was performed to identify differences in being at high risk of developing SBD (yes/no). Moreover, possible risk factors for the SDB were assessed by binary logistic regression. The *p*-value was set to <0.05 as statistically significant.

## 3. Results

A total of 1866 parents out of 2000 agreed to participate and complete the questionnaire (93.3%). The mean age of the participants was 9.67 ± 1.78 years. Among all the participants, a total of 1027 children were female (55%) and 839 were male (45%). The younger group and older group consisted of a total of 44.2% and 55.8% of participants, respectively. The outcome of the PSQ scoring among all the participants was presented in Table 1. Based on the PSQ scores, a total of 243 children (13%) were categorized as high risk of SBD.

Table 2 showed that the most prevalent symptom of SBD is mouth breathing (14.4%) and the least prevalent symptom is witnessed apnea (6.6%). The Chi-square test showed that there were significant differences between gender (females/males) and age (younger/males) in relation to being at high risk of developing SBD (*p* < 0.05). Moreover, there was a significant difference between high and low-risk children concerning habitual snoring, mouth breathing, witnessed apnea, being overweight, and bedwetting (Table 2).

Table 3 presented that binary logistic regression exhibited no significant association with gender in terms of developing SDB. However, the younger age group is significantly associated with the risk of developing SDB. The risks of developing SDB were 1.43 times higher in younger children compared to older children. In addition, children with habitual snoring, witnessed apnea, mouth breathing, being overweight, and bedwetting were at 8.9 times, 2.15 times, 6.6 times, 4.57 times, and 4.81 times higher risk of developing SDB, respectively (Table 3).

## 4. Discussion

The current study determined the incidence of SDB and identified related risk factors among primary school children in Al-Kharj, Saudi Arabia with the Arabic version of PSQ, which is the most used pediatric sleep questionnaire [17,27]. The original PSQ was translated into the Arabic language by Baidas et al. (2018), which assessed 1350 Saudi children, with 91% of the questionnaire reporting good concordance [17].

The risk of undiagnosed and untreated pediatric OSA could result in significant medical comorbidities including, but not limited to, cardiovascular, cognitive, metabolic, and growth hormone dysfunction [29]. The lack of awareness of pediatric SDB is one of the main barriers for families to seeking proper care, which starts with a formal diagnosis by a sleep physician using PSG followed by proper treatment [1]. This is the first population-based study in Al-Kharj which determined the incidence of SDB among school-going children. The response rate of the current study is 93.3% from a total of 1866 participants which is relatively large compared to previous studies [17,30], and somewhat comparable to a recent Saudi study [13]. Such a high response rate (93.3%) might be attributed to the way the questionnaire was distributed through school principals who are figures of authority at their schools.

The prevalence of SDB varies by definition and ranges from 4–17% [1,7,8,9,10]. The main finding of the study showed that 13% of participants with an age limit of 6–12 years were at a high risk of SDB. Additionally, 14.4% were considered mouth breathing, and 6.6% had witnessed OSA. In comparison to previous Saudi studies that used PSQ and included children with similar age ranges and comparable gender distribution exhibited 21% and 23% of high-risk SBD which is higher compared to the current study [17,18]. Moreover, samples from the previous studies reported participants with habitual snoring are more reported (10.7% and 15.9%) and witnessed apneas reported to a lesser extent (3.4% and 4%) compared to the present study [17,18]. The variation between our findings and Baidas et al. (2018) is due to differences in the operational definition of what constitutes habitual snoring. Moreover, Al Ehaideb et al. (2021) conducted a PSQ-based survey study of 285 Saudi children seeking orthodontic treatment and found that 47.7% were at high risk of developing SDB [31]. They also reported 11.3% and 11.6% of their sample to have habitual snoring and witnessed apnea, respectively [31]. These larger numbers could be because their sample was collected from an orthodontic clinic at a tertiary public hospital that receives referrals of cases with moderate or severe forms of malocclusions. Globally, studies that used the PSQ reported the prevalence of children at high risk of developing SDB to range from 7.9% to 12.8% [1,32,33].

The current study shows that there is a significant difference (*p* = 0.009) in SBD risk factors between younger and older groups. Additionally, it also showed that younger children were 1.43 times more likely to be at high risk of developing SDB compared to older children. In children, enlarged adenoids are the main reason for developing SDB and adenoidectomy/tonsillectomy is considering the primary treatment of SDB in children [9,34]. Adenoids reach their maximum size between the ages of 5 and 7 and begin to shrink afterward [35]. Therefore, enlarged adenoids were more likely to be found in the younger age group in our sample aged 6 to 9 years, and this might be the reason why the younger age group exhibited increased risks of developing SDB. In addition to enlarged adenoids, obesity has also been considered a possible cause of OSA. This study showed that there was a significant association between the high-risk group and children being overweight as perceived by their parents, which agrees with other studies conducted in Saudi Arabia [17,18,31].

In terms of gender distribution, this study showed a significant association between females and OSA, which is opposite to the reported male predilection of pediatric OSA in other studies [1,11,12,14,17]. The reason why there are usually more males affected by OSA is suggested to be due to the differences in the puberty age between males and females, as females enter puberty first. This variation in OSA prevalence between males and females usually increases as they age [1]. This finding reinforced the information from the previous studies where a significant difference between gender and OSA was observed, including children older than 12–13 years [36,37], while studies that did not show a significant association between gender and OSA were mostly limited to the younger age group [1,10]. This might explain why the current study, which was limited to children younger than 12 years old, was not in line with other studies in terms of the association between males and OSA. Nonetheless, it is important to note that the unique contribution of gender in the regression model was not significant in relation to being at high risks of developing SBD, controlling for other variables (including age, habitual snoring, witnessed apnea, parent perception of the child being overweight, and bedwetting). Therefore, cautious interpretation is warranted regarding this study findings in terms of gender association with, and contribution to, being at high risk of developing SBD.

It is not surprising that this study showed a significant association between high risks of developing SBD and snoring as snoring is one of the main symptoms of SDB and OSA [1]. Similar findings were observed in other previous studies [6,17,31]. Hence, parents need to appreciate the importance of seeking medical care when their child snores during sleeping. In the future, it would be worthwhile to explore whether the parents who participated in the study have considered medical care for their children, especially those with a high risk of pediatric SDB.

One of the common misapplications of PSQ questionnaires in the literature is the interpretation of a “yes” response to question no. 6 in the first domain (Table 1), as many authors have interpreted this as the presence of OSA [17,29]. It is recommended that authors use the term “witnessed apnea” for “yes” responses to this question instead of the inaccurate assumption that the child has OSA or is considered at high risk to develop SDB. The current study also showed that the high-risk group (19.3%) is more associated with OSA compared to the low-risk group (4.7%). Additionally, children with witnessed apneas were 2.84 times more likely to be at high risk for SDB. However, not every child with witnessed apnea was at high risk for SDB. There were 77 children who had witnessed apneas yet were considered low risk for SDB based on the answers to the remaining questions.

This study consists of some risk of bias due to the nature of its methodology. Moreover, being overweight was assessed by the parent’s perception. Using the body mass index would have been more objective yet was much harder to do especially given the large sample used in the current study. Additionally, having a PSG would have given a definitive OSA diagnosis. A future study could include a subgroup of the total sample to assess the prevalence more accurately. Although it is not the aim of the study, public awareness programs regarding snoring for children and the risk of pediatric SDB and its symptoms need to be implemented in schools as many local studies have reported similar findings of high rates of snoring and witnessed apnea.

## 5. Conclusions

In conclusion, 13% of the school-going children in Al-Kharj Saudi Arabia are at high risk of developing SDB at a younger age. Moreover, habitual snoring, mouth breathing, being overweight, bedwetting, and witnessed apnea were more prevalent in children with a high risk of SDB. Thus, the importance of further exploration of SDB among Saudi school-going children needs to be recognized, strategized, and materialized.

## Figures and Tables

**Table 1 healthcare-11-00747-t001:** Affirmative responses to PSQ questions including frequency and percentages.

Domain	Question	*n* (%)
Snoring/breathing problems	Snores more than half the time during sleep	202 (10.8)
	Always snores during sleep	167 (8.9)
	Snores loudly	199 (10.7)
	Heavy or loud breathing	230 (12.3)
	Difficulty in breathing	159 (8.5)
	Has stopped breathing during sleep	124 (6.6)
	Mouth breathing during the day	268 (14.4)
	Dry mouth upon waking	304 (16.3)
	Wets bed, walks during sleep or wakes up scared during the night	209 (11.2)
Daytime sleepiness and development	Wakes up unrefreshed	447 (24.0)
	Wakes up with a headache	180 (9.6)
	Difficult to wake the child up	435 (23.3)
	Sleepiness during the day	340 (18.2)
	Sleepiness during the day noticed by the teacher	179 (9.6)
	Has stopped growing at a normal rate	122 (6.5)
	Parent perception of the child being overweight	219 (11.7)
Inattention/hyperactivity	Does not respond quickly when spoken to	297 (15.9)
	Difficulty in organizing and managing tasks	328 (17.6)
	Easily distracted by external stimuli	435 (23.3)
	Seems restless and moves when seated	234 (12.5)
	Looks in a hurry all the time	761 (40.8)
	Interrupts others during speech	369 (19.8)
Number of children at high risk of sleep-disordered breathing (eight or more yes responses)		243 (13)

*n*; Total number, %; percentage.

**Table 2 healthcare-11-00747-t002:** Chi-square test results of differences in demographic variables and SDB symptoms related to SDB risk.

Variable			All Children	Low Risk	High Risk	*p*-Value
			*n* (%)	*n* (%)	*n* (%)	
Personal Characteristics	Sex	Male	839 (45.0)	745 (45.9)	94 (38.7)	0.035
		Female	1027 (55.0)	878 (54.1)	149 (61.3)	
	Age group, years	6–9	825 (44.2)	700 (43.1)	125 (51.4)	0.015
		10–12	1041 (55.8)	923 (56.9)	118 (48.6)	
SDB symptoms	Habitual snoring	Yes	167 (8.9)	101 (6.2)	66 (27.2)	<0.001 *
	Witnessed apnea	Yes	124 (6.6)	77 (4.7)	47 (19.3)	<0.001 *
	Mouth breathing	Yes	268 (14.4)	145 (8.9)	123 (50.6)	<0.001 *
	Parent perception of the child being overweight	Yes	219 (11.7)	139 (8.6)	80 (32.9)	<0.001 *
	Bedwetting	Yes	209 (11.2)	127 (7.8)	82 (33.7)	<0.001 *

SDB: sleep-disordered breathing, *n*; total number, %; percentage, *; significant difference (<0.05).

**Table 3 healthcare-11-00747-t003:** Binary logistic regression analysis of sleep-disordered breathing risk.

Variables	Odds Ratio	95% Confidence Interval		*p*-Value
		Upper	Lower	
Gender (female)	1.40	−0.032	−0.001	0.76
Age (younger)	1.43	−0.045	−0.014	<0.001 *
Snoring	8.90	0.147	0.174	<0.001 *
Mouth breathing	6.60	0.121	0.149	<0.001 *
Witnessed apnea	2.15	0.051	0.081	<0.001 *
Parent perception of the child being overweight	4.57	0.071	0.101	<0.001 *
Bedwetting	4.81	0.098	0.127	<0.001 *

*; significant difference (<0.05), %; percentage. Dependent variable: risk for sleep-disordered breathing (high or low).

## Data Availability

Not applicable.

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
