# Peer review of "Sleep-Disordered Breathing among Saudi Primary School Children: Incidence and Risk Factors"

_healthcare, 2023, doi:10.3390/healthcare11050747_

Round 1
Reviewer 1 Report
Please see the attachment

Reviewer 2 Report
Dear authors,
the manuscript "Sleep-Disordering Breathing between Saudi Primary School Children in Mixed Dentition: Incidence and Risk Factors" deals with a topic that has aroused much interest in recent years and requires synergistic insights between different branches of medicine. The analysis was conducted on the basis of a well-structured and reproducibly described research design. The information contained in the introduction needs to be integrated with further studies of the literature. The data has been correctly analyzed and commented. The conclusions are consistent with the results obtained. The limits of the study are not negligible, however I believe that this preliminary study can lead to an advancement of current knowledge as a basis for further future investigations.
The English language is fluent, with the exception of some grammatical and typo errors.
Best regards
Author Response
REVIEWER 2
Dear authors, the manuscript "Sleep-Disordering Breathing between Saudi Primary School Children in Mixed Dentition: Incidence and Risk Factors" deals with a topic that has aroused much interest in recent years and requires synergistic insights between different branches of medicine. The analysis was conducted on the basis of a well-structured and reproducibly described research design. The information contained in the introduction needs to be integrated with further studies of the literature. The data has been correctly analyzed and commented. The conclusions are consistent with the results obtained. The limits of the study are not negligible, however I believe that this preliminary study can lead to an advancement of current knowledge as a basis for further future investigations.
The English language is fluent, with the exception of some grammatical and typo errors.
Best regards
Reply to reviewer: Thank you so much for your comment. We have added information as per your comments.
Reviewer 3 Report
Dear Authors
the paper is well structured even if it is based only on questionnaries: as you know OSAS needs a clinic and instrumental diagnosis too.
The references are too many, please delete the oldest ones. Some minor errors in English are present.
Author Response
REVIEWER 3
Dear Authors
The paper is well structured even if it is based only on questionnaries: as you know OSAS needs a clinic and instrumental diagnosis too.
The references are too many, please delete the oldest ones. Some minor errors in English are present.
Reply to reviewer: Thank you so much for your comment. The corrections have been done as per your comments.
Reviewer 4 Report
The authors present a study aimed to identify the incidence and risk factors of SDB using an Arabic version of the pediatric sleep questionnaire (PSQ). The participants were divided into two groups (younger group: 6-9 years and older group: 10-12 21 years). 22 which 44.2% were from the younger group and 55.8% were from the older group. 13% of children were suffering from a high risk of SDB. Chi-square test and logistic regression analyses within this study cohort showed a significant association between SDB symptoms (habitual snoring, witnessed apnea, mouth breathing, being overweight, and bedwetting) and risk of developing SDB.
This study is interesting. However, there are a few major concerns.
1. Highlight how this study adds to the current knowledge
2. Validity and reliability compared to the gold standard test were not performed. Please justify this.
3. Subjective study results in recall bias and parental bias too. How did the authors address this issue?
Author Response
REVIEWER 4
The authors present a study aimed to identify the incidence and risk factors of SDB using an Arabic version of the pediatric sleep questionnaire (PSQ). The participants were divided into two groups (younger group: 6-9 years and older group: 10-12 21 years). 22 which 44.2% were from the younger group and 55.8% were from the older group. 13% of children were suffering from a high risk of SDB. Chi-square test and logistic regression analyses within this study cohort showed a significant association between SDB symptoms (habitual snoring, witnessed apnea, mouth breathing, being overweight, and bedwetting) and risk of developing SDB.
This study is interesting. However, there are a few major concerns.
- Highlight how this study adds to the current knowledge
- Validity and reliability compared to the gold standard test were not performed. Please justify this.
- Subjective study results in recall bias and parental bias too. How did the authors address this issue?
Reply to reviewer: Thank you so much for your comment. The correction has been done as per comments.
Reviewer 5 Report
As the authors mention within the limiting methodology factors of the study, this is a research which is purely based on a questionnaire, on the subjective interpretation of the parents regarding sleeping patterns and contains no clinical data regarding the patients enrolled, not even proper assessment of normal weight/overweight/obesity for the patients enrolled, which plays a very important role in the development of sleep-disordered breathing.
The title has nothing to do with the objective and purposes on the study, there is no mention regarding the dentition of children enrolled, nor how it might influence sleep-disordered breathing.
Author Response
REVIEWER 5
As the authors mention within the limiting methodology factors of the study, this is a research which is purely based on a questionnaire, on the subjective interpretation of the parents regarding sleeping patterns and contains no clinical data regarding the patients enrolled, not even proper assessment of normal weight/overweight/obesity for the patients enrolled, which plays a very important role in the development of sleep-disordered breathing.
The title has nothing to do with the objective and purposes on the study, there is no mention regarding the dentition of children enrolled, nor how it might influence sleep-disordered breathing.
Reply to reviewer: Thank you so much for your comment. The correction has been done as per comments.
Round 2
Reviewer 5 Report
I stand by my previous comments.
Author Response
Reviewers comments
As the authors mention within the limiting methodology factors of the study, this is a research which is purely based on a questionnaire, on the subjective interpretation of the parents regarding sleeping patterns and contains no clinical data regarding the patients enrolled, not even proper assessment of normal weight/overweight/obesity for the patients enrolled, which plays a very important role in the development of sleep-disordered breathing.
The title has nothing to do with the objective and purposes on the study, there is no mention regarding the dentition of children enrolled, nor how it might influence sleep-disordered breathing.
Authors’ response
Thank you for your valuable feedback.
The study methodology did not objectively include/exclude children based on predetermined/assessed criteria for several reasons. Objective evaluation of dentition/malocclusion features and sleep patterns that are associated with SDB require additional assessment tools (cephalometric radiograph and nocturnal sleep-based polysomnography, respectively) rather than clinical examination alone, and the use of these tools was beyond the scope of this study. Regarding proper assessment of the children’s weight, the questionnaire included a question on whether parents perceived their children being overweight and we agree it is a limitation of the study as we mentioned in our discussion.
In addition to the above-mentioned points, and because our study aim is more descriptive than inferential, the interpretation of the study’s findings, discussion and conclusion are confined to its aim, design, and approach, while recognizing and explicitly addressing its limitations. Such limitations have been addressed and some are highlighted as a promising area for further research within the community. As opposed to analytical/experimental epidemiolocal studies, and in accordance with our study aim, we conducted a cross-sectional descriptive/observational epidemiological study to document the incidence of SDB and identify associated risk factors in an un-examined community rather than to test a hypothesis (i.e., associations between specific malocclusion and SDB or the reliability of PSQ findings in comparison to PSG results).
